# Growing Season Harvests of Shrub Willow (*Salix* spp.) Have Higher Nutrient Removals and Lower Yields Compared to Dormant-Season Harvests

Daniel P. De Souza 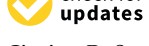, Mark H. Eisenbies * and Timothy A. Volk

College of Environmental Science and Forestry, State University of New York, Syracuse, NY 13244, USA
* Correspondence: mheisenb@esf.edu; Tel.: +1-315-470-4850

**Abstract:** The commercial establishment of shrub willow (*Salix* spp.) biomass crops with three- or four-year harvest cycles raises concerns about nutrient removal (NR). In addition, leaf-on harvests outside of the typical harvesting window are becoming more prevalent with a changing climate, and require a better understanding of the potential impact of these changes on biomass production and NR. This study examined the time of harvest effects for six harvest dates on the nutrient and biomass removal of four shrub willow cultivars in central New York State. There were significant differences in biomass in the first-rotation harvest; yields ranged between 77 and 85 Mg ha$^{-1}$ for the time of harvest treatments during the growing season, and between 93 and 104 Mg ha$^{-1}$ after dormancy. Harvest timing had significant effects on N and K removal in the combined wood and foliar biomass. Willow harvested in October removed comparatively higher amounts of N (77.1 kg ha$^{-1}$ year$^{-1}$) and P (11.2 kg ha$^{-1}$ year$^{-1}$) than other harvests. Potassium removal was greater for plants harvested in June (51.2 kg ha$^{-1}$ year$^{-1}$) and August (52.5 kg ha$^{-1}$ year$^{-1}$). Harvest timing and cultivar interactions suggest that targeted cultivar selection and deployment could maintain yields and limit excess nutrient losses.

**Keywords:** short rotation woody crops; short rotation coppice; *Salix*; harvest strategy; biomass production; nutrient management; leaf-on harvest

## 1. Introduction

Short rotation woody crops (SRWC), including shrub willow (*Salix* spp.), are considered potential biomass feedstocks to replace the fossil fuels that dominate the energy supply in the U.S. [1,2]. The initial establishment of willow crops at commercial scales has occurred in the U.S. [3,4]. These deployments have been facilitated by research and improvements in the system through the development of high yielding cultivars, improvements in harvesting systems, implementation of incentive programs, demonstration of environmental services and alternative applications, and opportunities to promote biodiversity and produce bioenergy [5–10]. Despite their potential and the improvements that have been achieved, a robust market for willow biomass crops has not developed. A reliable supply and demand for biomass must be guaranteed to support such a self-sustaining and reliable market. In the meantime, research related to shrub willow management harvesting impacts should continue in order to ensure that shrub willow producers can sustainably deliver biomass to end users year round.

Concerns about the sustainability of intensively managed forests and their respective nutrient demands have existed for decades [11–13]. Obvious comparisons and concerns have extended to SRWC [14]. The harvesting of woody biomass in forests or SRWC removes nutrients from the site; if they are not replaced by fertilization or natural mineralization, soil fertility may be impacted on sensitive sites and affect the long-term site productivity [13,15,16]. In addition, removal of material may disrupt crucial soil biological and ecological functions [17]. There are numerous strategies employed to reduce nutrient

losses during forest biomass harvest operations: (a) retain adequate quantities of slash (coarse and fine woody debris) on-site; (b) retain or leave tree foliage on-site to retain nutrients; and (c) replace removed nutrients by fertilizing biomass sites with wood-ash or other sources [18]. Existing cut-and-chip harvest operations in short rotation willow leave from 2.5 to 6.1 Mg ha$^{-1}$ on 95% of the area within a site, which generally accounts for 6 to 22% of the harvested biomass [19–21], although these values can be higher under some conditions [19,22]. Regardless of the amounts, the predominant recommendation for harvesting shrub willow is that it should occur after leaf fall and before bud set to reduce nutrient removal (NR) at harvest [23]. This timing allows some nutrients to be translocated from the leaves to the root system and stem, or returned to the soil as litter fall [1].

In practice, commercial harvests have not necessarily occurred within the recommended window because site conditions, especially on poorly drained marginal land in the northeast U.S., may limit machine access during the dormant season and operators may elect to expand the harvesting window [24]. The timing of willow harvests can potentially impact coppice regeneration and regrowth if harvested during the growing season [25–27], and does impact harvester throughput and fuel use [24] and as a result will impact production costs [19,28]. The effect of the timing of harvest in coppice systems has arisen as a key research question and may have different implications across regions, and among SRWC species including shrub willow cultivars [29–33]. However, the phenotypic variation in coppicing relative to harvest timing is variable [29,34]. Dormant-season harvest is recommended to ensure maximum sprout vigor, compared to growing-season harvest, given the higher availability of carbohydrate reserves in roots after leaf fall, which will support the initial growth of new sprouts after harvest [25,30]. In addition, nutrient translocation within the plant relative to season could be impactful on harvest timing in commercial willow stands [35,36]. The initial growth rate of shrub willow coppiced stems is very dependent on solar radiation, temperature, and water availability; however, studies have shown that the vigor of sprouts and the sprouting ability decreased severely when plants were harvested during an actively growing stage [29,30,37].

While most existing guidelines for shrub willow suggest harvests should occur when trees are dormant [18,23], there is not much information about the impact on productivity and NR when harvest is conducted during the growing season. Studies have been conducted looking at nutrient cycling in willow foliage and related to seasonal harvests under specific conditions [25–27,35]; however, the direct application of these results for developing harvest schedules in the Northeastern U.S. for commercial willow is not clear. Additionally, nutrient translocation and storage is variable between cultivars [38]. In practice, harvesting schedules are difficult to predict because they are subject to ground conditions, weather, and machine availability, which may delay, hinder, or even preclude activities at an optimal time for the crop. Shrub willow crops in central New York State (NY) are frequently planted on marginal agricultural land, which is generally due to poor site drainage as opposed to fertility issues. Another key aspect of dormant-season harvests in NY is that the weight of machines is better supported on frozen ground, which enhances mobility and prevents soil displacement [18,23]. However, temperatures of sufficient duration to freeze the ground in central and northern NY have become increasingly unreliable. The extended freezes that are required often occur after the first significant snowfalls, which insulate the ground and prevent it from hardening to a sufficient depth for equipment. Commercial willow growers have responded to these changing circumstances by initiating their harvest operations as early as mid-August to ensure access to the scheduled acreage. In principle, leaf-on harvests remove additional above ground biomass and nutrients, which could impact the long-term productivity of the site.

Given the operational realities and importance of nutrient management strategies that ensure the system's production over multiple coppice cycles, the objective of this project is to evaluate the effects of the harvest timing on NR and above ground biomass production for four cultivars of shrub willow crops in central NY. Specifically, the harvest timing affects the back end of the previous cutting cycle by removing plants before they have completed

the full growing season, reducing yields and increasing NR. Harvest timing affects the front end of the subsequent cutting cycle when plants resprout, but if harvesting occurs late in the growing season the new growth may not harden off and die back over the winter and will not contribute the total biomass in the next harvest. As such, the hypothesis is that, first, the time of year that a harvest occurs will affect nutrient removal based on the amount and type of plant materials that are removed with the harvest. Second, harvest timing will affect subsequent regrowth. Third, differences may be cultivar dependent.

## 2. Materials and Methods

A willow stand located in Canastota, NY (43°03′05″ N, 075°44′19″ W) was established in the spring of 2002. The soils at the site are classified as Cazenovia and Camillus silt loams (fine-loamy, mixed, active, mesic Oxyaquic Hapludalfs and fine-loamy, mixed, active, mesic Typic Eutrudepts) that are moderately well drained with a depth to water table ranging from 61 to 121 cm, and a depth to bedrock of more than 200 cm [39,40]. The climate is temperate humid and cold, with a mean annual precipitation ranging from 973 to 1017 mm [41]. Following guidelines at the time, a suite of six cultivars (Table 1) were planted in monoclonal blocks in double rows with a spacing of 1.5 m between double rows, 0.76 m within double rows, and 0.6 m between plants (14,400 plants ha$^{-1}$). The crop was coppiced after the 2002 growing season, to induce the growth and development of multiple sprouts per stool, and urea applied at a rate of 100 kg N ha$^{-1}$ at the beginning of the second growing season.

**Table 1.** Shrub willow cultivars included in the time of harvest study in Canastota, NY. The trial originally had 6 cultivars, but two were removed after the first harvest due to low survival.

| Cultivar ID * | Diversity Group | Species/Pedigree | Note |
| --- | --- | --- | --- |
| 9882-25 | PUR | *S. purpurea* L. | narrower strips |
| 9870-40 | MIYA | *S. miyabeana* Seem. | |
| 9871-41 | MIYA | *S. miyabeana* Seem. | |
| 95311 | ERIO | *S. eriocephala* Michx. | low survival, removed |
| SX61 | MIYA | *S. miyabeana* Seem. | low survival, removed |
| SX67 | MIYA | *S. miyabeana* Seem. | |

* Cultivars with numbered ID are from State University of New York College of Environmental Science and Forestry (ESF) breeding program. First two numbers indicate year cross was made (i.e., 98 = 1998) and remaining numbers indicate family and individual in the family. SX61 and SX67 were included because of their consistent performance in previous trials.

Time-of-harvest treatment (ToH) plots that included harvests at four stages of plant development during the growing season and two dormant-season harvests were installed in each cultivar block using a strip-strip factorial design (Figure 1, Table 2). Plots consisted of five double rows in width (11.4 m) and length of 14 plants per double row (4.3 m) for a total area of 49 m$^2$. Due to insufficient rows on the westmost block (cultivar 9882-25), these ToH plots were narrowed to four double rows (9.3 m) and 40 m$^2$ in area. Due to poor survival following harvest, two cultivars were removed from the trial (cultivars 95311 and SX61; Table 1). Measurement plots were established within each ToH area consisting of 18 plants (center three double rows for 9870-40, 9871-41, and SX67 and two double rows in the 9882-25 plots). Hence, the measurement plots were buffered within the ToH plots by one double row on the east and west edges, three plants at the southern edge and one plant at the northern edge.

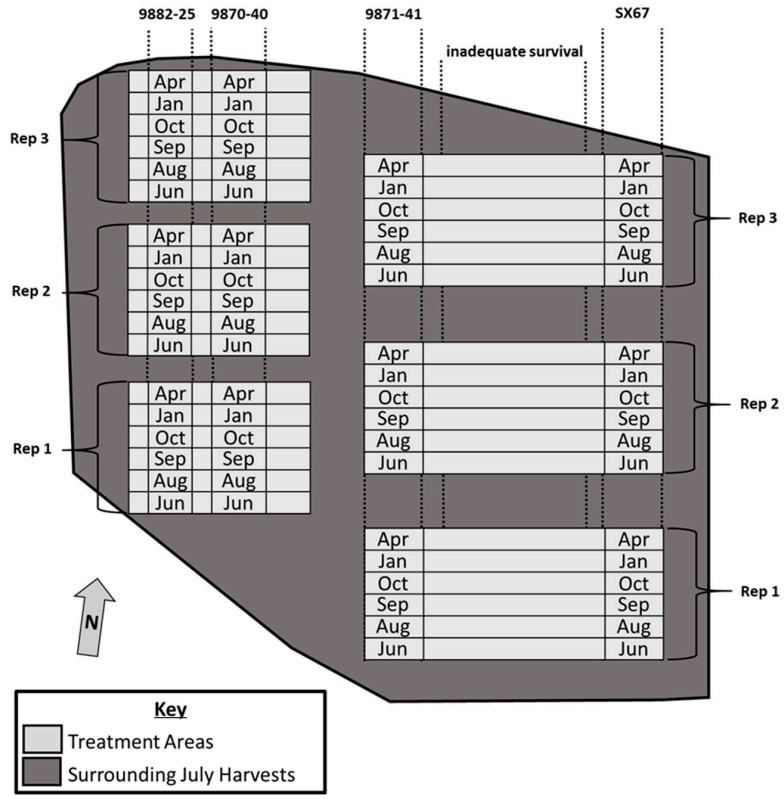

**Figure 1.** Schematic of the experiment with the four cultivars used in the study and layout of plots and replications. Dark grey area was harvested at the same time as the first harvest date (Jun). Grey areas indicate the time-of-harvest treatment (ToH) plots associated with the harvest dates, and areas that were treated but not included due to low survival.

**Table 2.** Time of harvest, age of aboveground plant parts in months, growing degree days (GDD) for each harvest date, and stage of plant growth used to study the impacts of timing of harvest on nutrient removal and coppice regrowth of shrub willow crops.

| Time of Harvest Treatment | Harvest Date | Plant Stage | Stem Age at First Harvest (Months) | GDD * for Entire First Rotation | Stem Age at Second Harvest in January 2011 (Months) | GDD * for Entire Second Rotation |
|---|---|---|---|---|---|---|
| June | 6–8 June 2007 | After full leaf out | 54 | 7187 | 54 | 6745 |
| August | 2–8 August 2007 | After bud set | 56 | 7739 | 52 | 6146 |
| September | 13–18 September 2007 | Starting senescence | 57 | 8164 | 51 | 5747 |
| October | 15–19 October 2007 | Mid-fall (leaves dropping) | 58 | 8366 | 50 | 5552 |
| January | 21–24 January 2008 | Dormant | 61 | 8417 | 47 | 5508 |
| April | 10–14 April 2008 | Before leaf out began | 64 | 8420 | 44 | 5460 |

* Calculated using 10 °C as base temperature.

Implementation of the time-of-harvest treatments started in June 2007 (root systems were entering their sixth year with stems entering their fifth growing season) at the south end of the field and progressed northward so that each cut would take place on the southern exposure of each replication (Figure 1). While this approach does release plants in subsequent plots, the purpose was to ensure equitable lighting for sprouts that emerge from willow stools soon after cutting. Additionally, the harvest plan dictated the stand surrounding the ToH areas was harvested mechanically in June at the same time as the first ToH plot to simulate clear-cut conditions, maximize light, and avoid extraneous and/or unequal shading effects on the remaining ToH areas. Thus, the ultimate experimental design consisted of a 6 × 4 ToH × cultivar (ToH × C), strip-strip factorial experiment, also known as a split block, with three replications (blocks due to potential differences in drainage). Phase 2 of the plan consisted of a leaf-off harvest of second rotation growth that took place at the same time for all plots, December 2011.

Time-of-harvest treatment areas were hand harvested with gas-powered brush saw on their prescribed dates (Table 2). Willow inside the measurement plot were cut 5 to 15 cm above the soil surface and all the above ground parts of the plants were collected (including bole, branches, twigs, bark, and leaves), bundled, and weighed (scales suspended from tractor bucket as a combined sample) to estimate wet biomass. To collect foliage samples and stem:foliage information, three stems were selected representing the range of diameters represented in the bundle (small, medium, and large, relative to the diameter range at each plot) and the foliage material (e.g., leaves) were separated from the stem material (e.g., bole, branches, twigs, and bark). These stem subsamples were subsequently chipped, weighed, and a 1 to 2 kg sample retained to determine gravimetric moisture content [42]. Leaf subsamples from leaf-on harvests, were weighed, bagged, and retained to determine gravimetric moisture content. Thus, leaf mass, stem mass, stem:leaf ratio, and moisture content could be estimated for each harvest date and scaled to a per hectare basis. The second harvest was conducted simultaneously for all plots in December 2011 using the same methodology for plot biomass that was used in 2007. Data were annualized in the first and second rotations by the number of months between the ToH treatment and the 2011 harvest (Table 2); for example, production from plots harvested in June 2007 was divided by 4.5 years, while production from plots harvested in April 2008 was divided by 5.3 years).

Nutritional analyses of the biomass were performed on a subsample from the chips collected from each plot during the harvest. A 300–400 g representative sample of dried chips were ground in a Willey Mill using a 40-mesh screen, and a 3–5 g subsample produced. Similarly, a 3–5 g sample of foliar material was also produced. Samples were sent to the Agricultural Analytical Services Lab at the Pennsylvania State University (College Station, PA, USA) for plant tissue total analysis to determine the nutrient concentration of the biomass components. As described by the lab, determination of total N was carried out through the micro-Kjeldahl method, while the determination of P and was performed through the microwave acid digestion method and inductively coupled plasma atomic emission spectrometry (ICP-AES). Nutrient concentrations for foliage and stems were multiplied by their harvested weights for each ToH and cultivar plot and scaled to kg ha$^{-1}$. Finally, the data were annualized by the number of months the plants were growing for each of the ToH treatments in both the first and second rotation (Table 2).

Biomass production in 2007, 2011, and NR in 2007 for each time-of-harvest treatment, was calculated for each measurement plot. Mixed models were evaluated for a strip-strip factorial using the GLIMMIX procedure (SAS version 9.4, Cary, NC, USA) to estimate the effects of the six timings of time-of-harvest treatments (ToH), cultivars (C), and ToHxC interactions on total biomass production, annual yield, NR, and crop survival. A Satterthwaite degrees of freedom approximation was used. Main factors, interactions, and contrasts were tested at a critical level α of 0.05. Planned contrasts consisted of testing the cultivar SX67 (a common high-performing cultivar, Table 1) against the other three cultivars within a harvest date and its own performance throughout the growing season

(June/August/September) vs. dormant season (October/January/April). Significance tests were conducted using the pairwise least square means (LS means) differences (PDIFF command) in the LSMEANS and contrasts using the LSMESTIMATE statements [43,44], although arithmetic means and standard errors are presented.

## 3. Results and Discussion

### 3.1. Time of Harvest Effect on Biomass and Nutrient Removal

Growing season harvests affect recovered biomass by cutting fast growing willow before its growing season is complete. Harvest timing affected yields, which increased significantly from 35 Mg ha$^{-1}$ to around 50 Mg ha$^{-1}$ through the growing season (Table 3; Figure 2 top). In the 2007 harvest, the mean annual increment through the growing season was between 7.3 and 10.6 Mg ha$^{-1}$ year$^{-1}$, and jumped substantially in the second rotation to a range of 11.3 to 17.0 Mg ha$^{-1}$ year$^{-1}$. This increase was consistent with the results reported by Sleight et al. [45], which showed that when yields after the first cutting cycle in willow stands are low, they have a propensity to increase in the subsequent cutting cycle; willow crops in this area are usually in place for up to seven cycles of coppice regrowth. Leaf fall occurred following the October harvest. Late-season harvests (October–April) yielded a total of 7.87 Mg ha$^{-1}$ more biomass than early season harvests (June–September) ($p = 0.0068$). Mean foliar biomass within the 2007 harvest season was 3.42 Mg ha$^{-1}$. There were no significant differences in foliar biomass June through October while leaves were on the trees. The stem to foliage ratio trended slightly upward from 11.3 to 13.7 as the growing season progressed, but was not significant ($p = 0.8742$).

Overall, observed ToHxC effects for stem biomass were not significant ($p = 0.3546$), but were also evaluated using the planned contrast for the SX67 cultivar. The trend of increased biomass through the progression of harvests was driven by the cultivars 9870-40 and 9871-41 (Figure 2, bottom). However, based on the contrast, there were no significant differences within the SX67 cultivar harvests regardless of whether they occurred leaf-on or leaf-off ($p = 0.3107$). Additionally, total biomass for the June/August harvest dates for SX67 were significantly higher than the other cultivars ($p = 0.0415$). The cultivars 9870-40, 9871-41, and SX67 belong to the MIYA species group, while 9882-25 belongs to *PUR* (Table 1). The overall implication of this result is that cultivar responses to expanding growing seasons may be variable; some tolerant to the practice, and others intolerant.

**Table 3.** Summary of analyses of variance for the effects of time of harvest, cultivar, and ToHxC on biomass production following first rotation time of harvest treatments. Main effects significance tested at $\alpha = 0.05$. DF is degrees of freedom.

| Biomass | Time of Harvest (ToH) | Cultivar (C) | ToHxC |
|---|---|---|---|
| | *p*-Values | | |
| 2007 Stem | <0.0001 | 0.2013 | 0.3546 |
| 2007 Foliage | 0.8868 | 0.0469 | 0.0794 |
| 2007 Total | 0.0068 | 0.2375 | 0.4023 |
| 2011 Annual yield | <0.0001 | 0.3233 | 0.1488 |
| Cumulative 2007–2011 | 0.0027 | 0.3038 | 0.2707 |
| DF | 5 | 3 | 15 |
| Denominator DF | 40 | 6 | 40 |

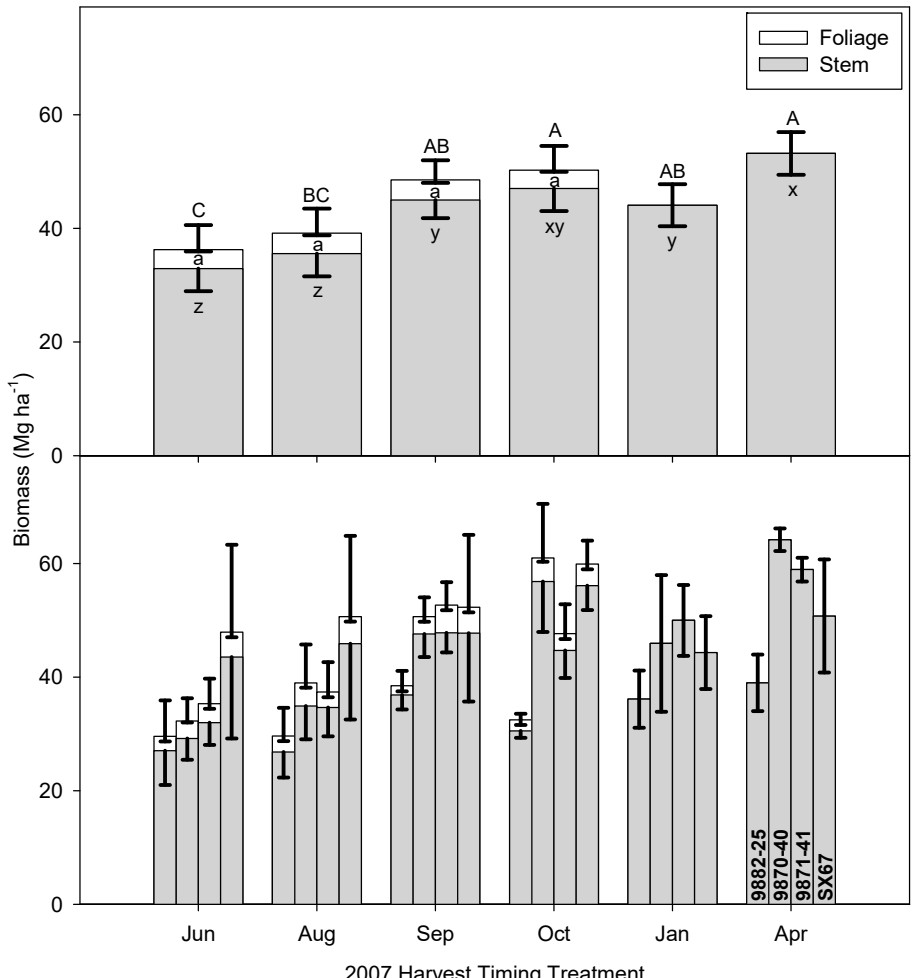

**Figure 2.** (**Top**) Mean willow stem and foliage biomass production for the first rotation time-of-harvest treatments (ToH) that occurred on the prescribed ToH areas between June 2007 and April 2008. Letters indicate significant differences between total biomass (upper case A-B-C), foliar biomass (lower case a-b-c), and stem biomass (lower case x-y-z). One-way error bar indicates the standard error for the arithmetic mean for total biomass (upward), downward in the white space for foliage, and downward in the gray space for stem components. (**Bottom**) Sets of bars showing stem and foliage biomass for individual cultivars as labeled on the rightmost group.

During the 2007 harvests, the ToHxC interaction for the combined stem and foliage and for the stem NR was significant for all nutrients except K at the alpha = 0.05 level (Table 4). Harvest date was significant for N, P, and K. While foliage had 3–10 times the concentration compared to stemwood (Table 5), NR was highest for the woody component due to quantity. Overall, the results suggest that the majority of losses are being driven by the total nutrient content of the stems, rather than foliage. Foliage is an undesirable component for some end users because it raises the ash and moisture content of the feedstock. Anecdotally, a lot of foliage is blown free from the biomass stream during mechanical harvesting operations, although the precise amount is not presently known, which may influence losses in commercial scale operations. Losses during harvesting were lower in this trial because measurement plots were hand harvested. Future equipment may use some on-site separation technology on the harvester (e.g., air classification) to reduce the amount of foliage in the material stream in an effort to improve feedstock quality [19].

**Table 4.** Summary of analyses of variance for the effects of time-of-harvest treatment (ToH), cultivar (C), and ToHxC on total nutrient removal (stem + foliage) on the 2007 harvest.

| Element | Time of Harvest (ToH) | Cultivar (C) | ToHxC |
|---|---|---|---|
| | *p*-Values | | |
| Combined stem and foliage | | | |
| N (kg ha$^{-1}$ year$^{-1}$) | 0.0293 | 0.0792 | 0.0249 |
| P (kg ha$^{-1}$ year$^{-1}$) | 0.0514 | 0.0879 | 0.0891 |
| K (kg ha$^{-1}$ year$^{-1}$) | 0.0002 | 0.0592 | 0.1043 |
| Stem | | | |
| N (kg ha$^{-1}$ year$^{-1}$) | <0.0001 | 0.0655 | 0.0062 |
| P (kg ha$^{-1}$ year$^{-1}$) | 0.0108 | 0.0717 | 0.0419 |
| K (kg ha$^{-1}$ year$^{-1}$) | 0.0724 | 0.0501 | 0.1331 |
| Foliage | | | |
| N (kg ha$^{-1}$ year$^{-1}$) | 0.0345 | 0.1233 | 0.1429 |
| P (kg ha$^{-1}$ year$^{-1}$) | 0.0076 | 0.1534 | 0.1646 |
| K (kg ha$^{-1}$ year$^{-1}$) | 0.0363 | 0.0589 | 0.1069 |
| DF | 5 | 3 | 15 |
| Denominator DF | 40 | 6 | 40 |

**Table 5.** Summary of wood and foliage nitrogen nutrient concentrations for time of harvest treatment main effects. Letters indicate significant differences within columns at the alpha = 0.05 level.

| Harvest Date | N | | P | | K | |
|---|---|---|---|---|---|---|
| | Wood | Foliage | Wood | Foliage | Wood | Foliage |
| | mg/kg | | | | | |
| June | 3.37 (0.18) C | 26.92 (0.70) A | 0.57 (0.04) | 3.43 (0.11) A | 3.18 (0.18) A | 14.44 (0.40) A |
| August | 3.53 (0.17) BC | 22.71 (0.64) B | 0.66 (0.04) | 2.04 (0.08) B | 3.42 (0.20) A | 12.12 (0.74) B |
| September | 3.55 (0.09) BC | 18.97 (0.68) C | 0.59 (0.03) | 1.61 (0.06) C | 2.63 (0.13) B | 9.38 (0.27) C |
| October | 4.12 (0.17) AB | 17.25 (0.51) C | 0.63 (0.03) | 1.70 (0.09) C | 2.21 (0.10) BC | 8.31 (0.78) C |
| Novembwer | 4.53 (0.50) A | | 0.60 (0.06) | | 2.36 (0.23) B | |
| April | 3.41 (0.26) C | | 0.49 (0.04) | | 1.88 (0.14) C | |

Nitrogen is generally one of the most limiting nutrients for most crops [11,46]. Nitrogen removal was most affected by season, but there were significant ToHxC interactions detected (Table 4), but were too erratic to be singled out on the figure (Figure 3). Significant effects of harvest date, cultivar, and ToHxC were observed on total N removal, with the highest removal observed by cultivar SX67 in January (110 kg N ha$^{-1}$ year$^{-1}$) and the lowest by 9870-40 in January (46 kg N ha$^{-1}$ year$^{-1}$). Removals by cultivar SX67 were significantly higher than the removals of all the other cultivars in January and statistically similar to cultivar 9870-40 in June and August. On the other hand, removals in September, October, and April were statistically similar for all cultivars. Overall, total N removal was lower in April (57 kg N ha$^{-1}$ year$^{-1}$) compared to October (78 kg N ha$^{-1}$ year$^{-1}$) and January (73 kg N ha$^{-1}$ year$^{-1}$). Total N removal for the other time-of-harvest treatments was not significantly different. When just considering woody biomass, the lowest removals were in January (44 kg N ha$^{-1}$ year$^{-1}$), August (45 kg N ha$^{-1}$ year$^{-1}$), and September (46 kg N ha$^{-1}$ year$^{-1}$).

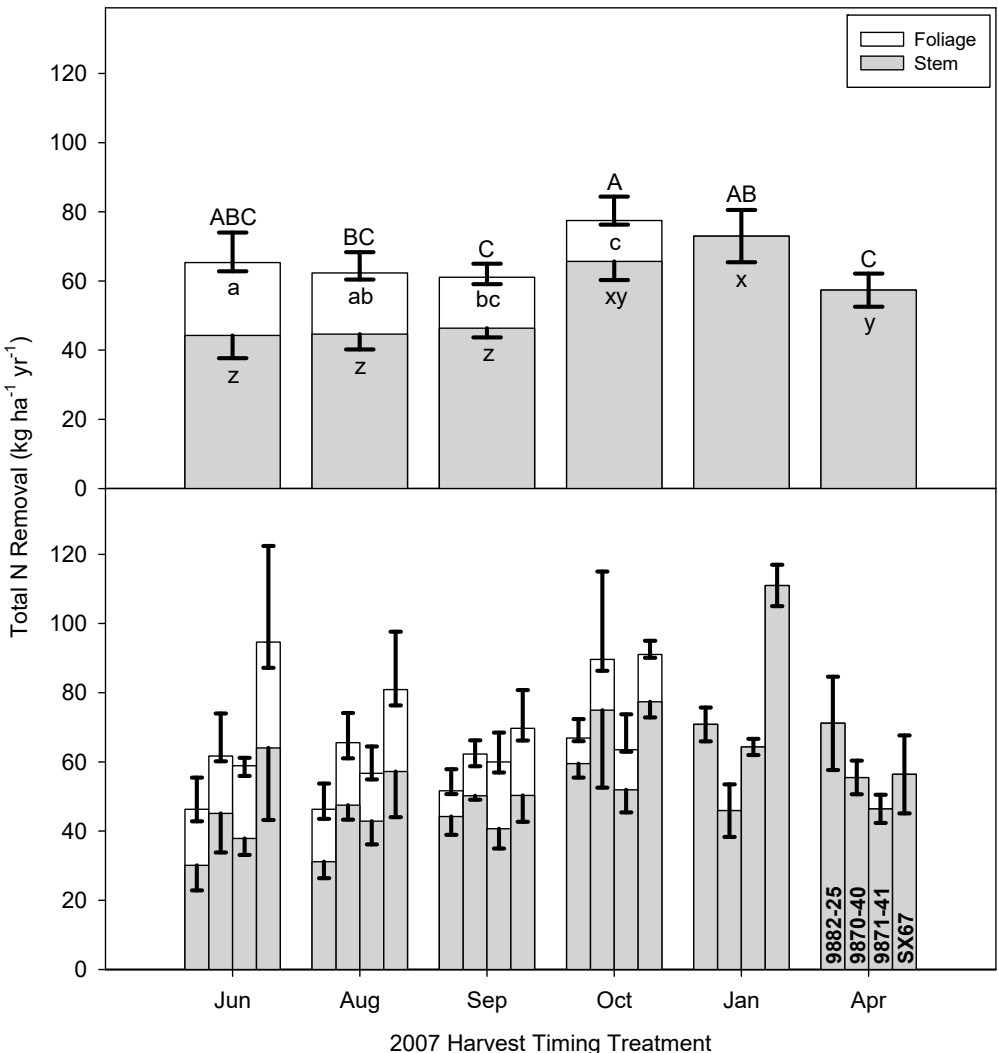

**Figure 3. (Top)** Nitrogen removal (mean ± SE) in the form of stem and foliage biomass for the first rotation time-of-harvest treatments (ToH) that occurred on the prescribed ToH areas between June 2007 and April 2008. Letters indicate significant differences between total removal (upper case A-B-C), foliar removal (lower case a-b-c), and stem removal (lower case x-y-z). One-way error bar indicates the standard error for the arithmetic mean for total biomass (upward), downward in the white space for foliage, and downward in the gray space for stem components. **(Bottom)** Sets of bars showing stem and foliage biomass for individual cultivars as labeled on the rightmost group.

Observing the response of individual cultivars, N removal was proportional to biomass removal for harvests occurring on a particular date (Figures 2 and 3). Although there was not a significant cultivar effect ($p$ = 0.0792), the mean removal for cultivars ranged from 57.7 to 83.9 kg N ha$^{-1}$ year$^{-1}$. As the growing season progressed into September, N removal decreased relative to biomass removal. As the willow plants hardened for winter, there was a spike in nitrogen removal in October, and with the signal diminishing again as the dormant season progressed. Analysis indicated there were no significant differences in total N removal within the July–September harvests, but there was some variability in the later Oct–Apr harvests ($p$ = 0.0293). In this hand harvested trial, the vast majority of the leaves were removed at the time of harvest and included in both the biomass and nutrient calculations. However, in large-scale harvests using single pass cut-and-chip harvesting systems only a fraction of the foliage is removed from the field. Observations over hundreds of hectares of large-scale harvesting [24] indicate that more foliage is lost when harvests occur in the fall since leaves are more easily knocked off the stems as the stems are cut

and shaken as they are pulled into the harvester. Less dense foliage is also returned to the site when material is blown from the harvester into wagons or other collection vehicles. In addition to foliage, studies show that 7 to 15% of the standing biomass in willow fields can be left behind during mechanical harvesting operations [19–21]. There is a lack of good data on the proportion of foliage returned to the site during mechanical harvesting, but it could impact nutrient dynamics in these systems and should be considered. The hand-harvested data in this trial is unique in the region, but samples will contain more foliage removal than occurs in mechanical harvests.

Based on the planned contrasts for the SX67 cultivar, there was a significant difference in N removal compared to the other cultivars ($p = 0.0012$), which was possibly driven by the total amount of biomass removed as there were no differences in N concentration in cultivar components ($p > 0.3750$). The differences were variable over the course of the time-of-harvest treatments ($p > 0.0465$); a trend is apparent in the foliar N for SX67, for example (Figure 3), but ultimately not significant ($p = 0.2663$). SX67 had stem and foliar N concentrations at the low end of the range among willow cultivars that were tested (Tharakan et al., 2004). In this study, this trend held with foliage but was not significant ($p = 0.1302$). However, yields in this trial were consistently high across all the harvest date, emphasizing the importance that total biomass removal has on NR. It is important to note that there are differences in the timing of leaf drop between the cultivars [47,48] and that this will influence the amount of nutrient removed during harvesting both in terms of biomass and the nutrient concentration of the remaining foliage.

The combined pattern for P was congruent with N in that P removal was proportional overall to biomass removal, but in the case of P, the combined removal was not significant relative to harvest timing or cultivar at an alpha = 0.05 level (Table 4, Figure 4). The range of means among the cultivars was 7.7 to 11.6 kg P ha$^{-1}$ year$^{-1}$. However, there were significant main effects when considering stem and foliage individually. Considering planned contrasts, total P removals in the early growing season by SX67 were 4.7 kg P ha$^{-1}$ year$^{-1}$ higher than other cultivars ($p = 0.0031$). Da Ros et al. [38] clearly shows the precedent for variability between cultivars in terms of nutrient translocation and storage. While there are differences in the patterns of P removal in wood and foliage, the total amounts removed are relatively small (7–14 kg P ha$^{-1}$ year$^{-1}$) and in the northeast U.S., where this trial took place, many of the soils have more than adequate or excessive amounts of P due to the inherent characteristics of this soil and its management, especially additions of manure and P fertilizers [49]. In other regions or on disturbed sites with lower soil P levels, the removal patterns seen here need to be considered in nutrient management decisions.

Potassium is among the most abundant inorganic chemicals in plant cellular media and is the second most abundant nutrient in leaf tissues after N. Potassium plays important roles in the development and functioning of plants. Potassium is mobile in plant tissues and organs and it cycles rapidly among plant parts [50]. The total removal of K was significantly affected by time of harvest ($p = 0.0002$, Table 3). Potassium removal decreased from a high of 52 kg K ha$^{-1}$ year$^{-1}$ in June/August to approximately 30 kg K ha$^{-1}$ year$^{-1}$ by the following April (Figure 5). Planned contrasts for SX67 suggest that the cultivar does not change significantly between the growing and dormant season ($p = 0.1043$). In contrast to the results of N and P, the removal of K via woody biomass was higher during the full-leaf out stages of the crop (40.0 kg K ha$^{-1}$ year$^{-1}$ in June and 42.6 kg K ha$^{-1}$ year$^{-1}$ in August), compared to the late-season harvests (September, October, January, and April), despite stem biomass removal being higher later in the season. Similarly, K removal via foliage was higher in the early season (>10.5 kg K ha$^{-1}$ year$^{-1}$ before June–September) and decreased as the crop approached dormancy (7.2 to 9.8 kg ha$^{-1}$ year$^{-1}$).

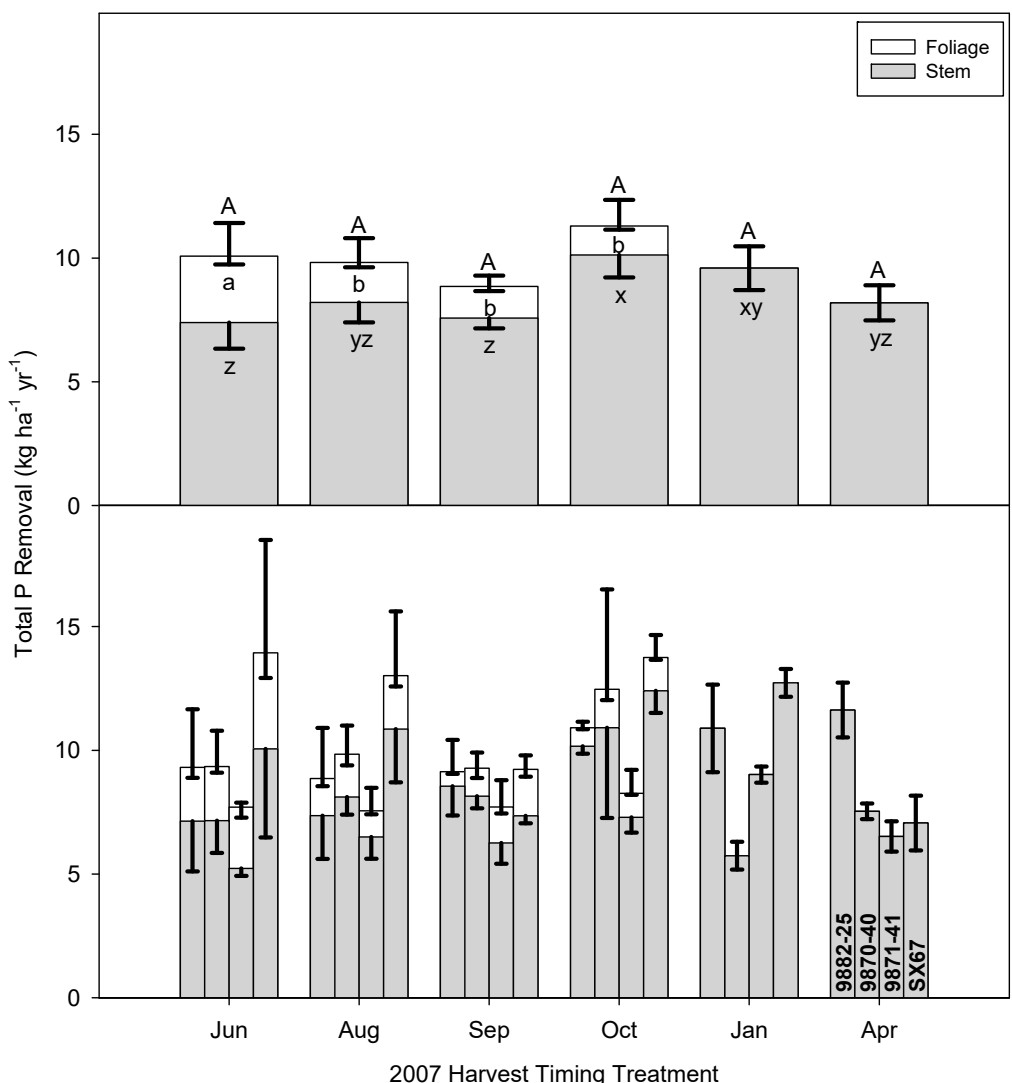

**Figure 4.** (**Top**) Phosphorus removal (mean ± SE) in the form of stem and foliage biomass for the first rotation time-of-harvest treatments (ToH) that occurred on the prescribed ToH areas between June 2007 and April 2008. Letters indicate significant differences between total removal (upper case A-B-C), foliar removal (lower case a-b-c), and stem removal (lower case x-y-z). One-way error bar indicates the standard error for the arithmetic mean for total biomass (upward), downward in the white space for foliage, and downward in the gray space for stem components. (**Bottom**) Sets of bars showing stem and foliage biomass for individual cultivars as labeled on the rightmost group.

### 3.2. Time of Harvest Effect on Biomass and Second Rotation Biomass

How willow production responds to an expanding harvest season is a core objective of this study. The 2011 harvest (second rotation) represented four growing seasons of growth, plus any growth that occurred after the early to mid-season harvests during the summer/fall of 2007. The mean annual yield in the second rotation ranged between 11.3 and 17.0 Mg ha$^{-1}$ year$^{-1}$ (Figure 6). The second rotation yield was 5.3 Mg ha$^{-1}$ year$^{-1}$ more than the first rotation across all cultivars and times of harvest ($p$ = 0.0068). Late-rotation harvests (October–April) produced an average of 4.2 Mg ha$^{-1}$ year$^{-1}$ more biomass in the second rotation than areas where an early rotation harvest took place (June–September). Based on planned contrasts, there was no significant difference in the SX67 performance related to ToH treatments through 2011 ($p$ = 0.5094) and SX67 plots harvested in June/August performed better in 2011 than any of the other cultivars ($p$ = 0.0010) (Figure 6).

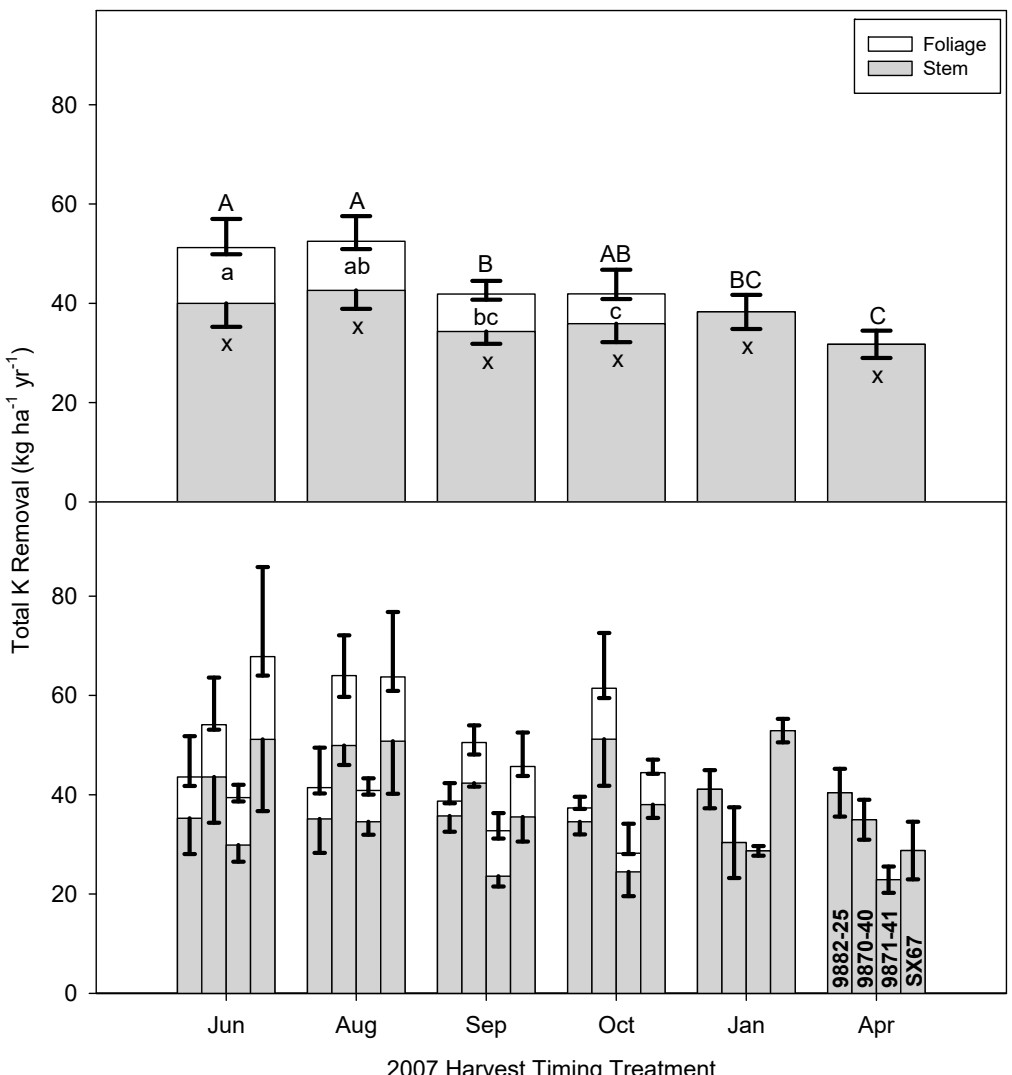

**Figure 5. (Top)** Potassium removal (mean ± SE) in the form of stem and foliage biomass for the first rotation time-of-harvest treatments (ToH) that occurred on the prescribed ToH areas between June 2007 and April 2008. Letters indicate significant differences between total removal (upper case A-B-C), foliar removal (lower case a-b-c), and stem removal (lower case x-y-z). One-way error bar indicates the standard error for the arithmetic mean for total biomass (upward), downward in the white space for foliage, and downward in the gray space for stem components. **(Bottom)** Sets of bars showing stem and foliage biomass for individual cultivars as labeled on the rightmost group.

It is common knowledge that yield performance varies considerably between willow cultivars. The differences in growth behavior between the cultivars within the growing season seems to be a unique result. What is particularly interesting is that these results suggest that certain cultivars, in this case SX67, although it has higher variability, its higher mean suggests it may be more resilient to the expanded harvest seasons that may be necessary to support a large-scale commercial willow system. Regardless of when SX67 was harvested in the first rotation, second rotation yields were high and fairly consistent. Both high yield and a tolerance for early season harvesting may be the necessary selection criteria for cultivar deployment. However, matching cultivars to favorable sites can be a challenge; the purpose of this paper is not to promote specific cultivars.

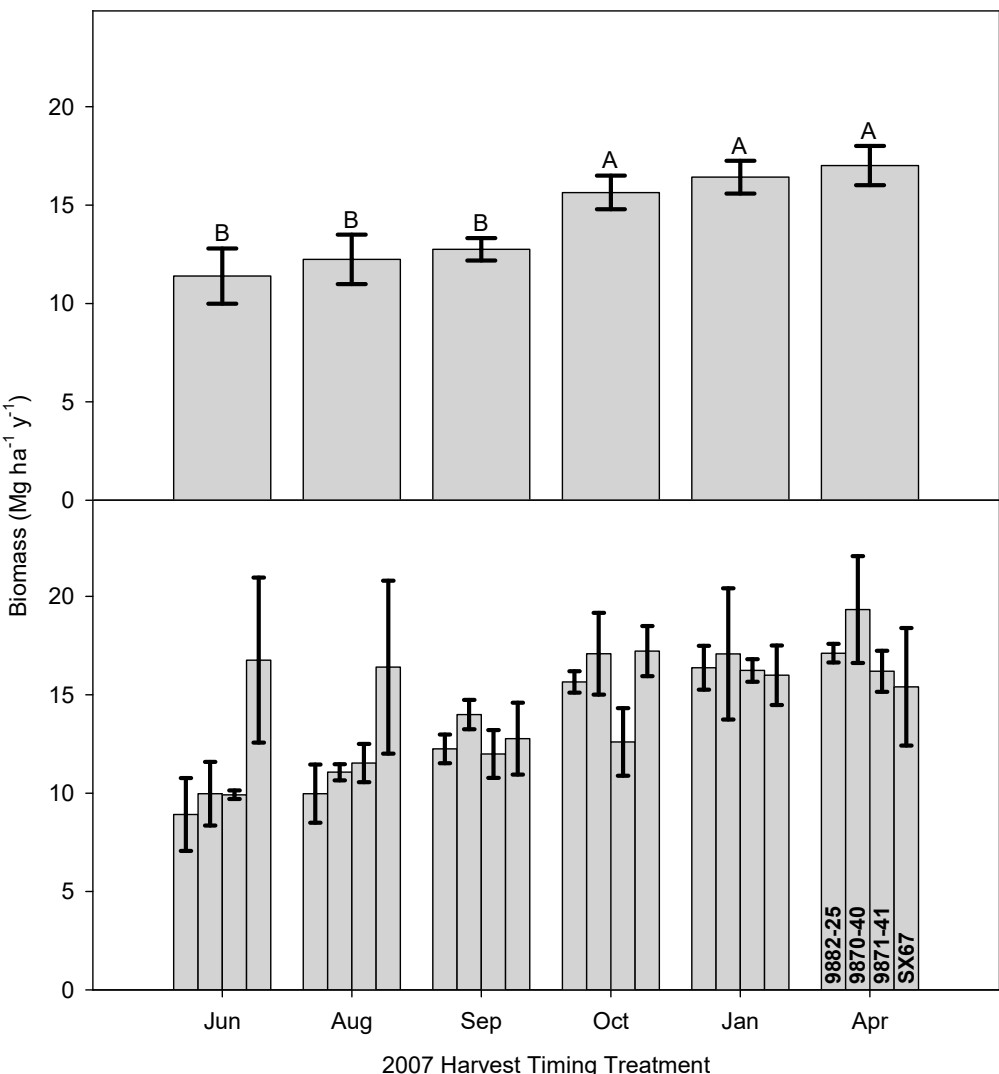

**Figure 6. (Top)** Mean and standard error of annual yield for the leaf-off, second rotation harvest in December 2011 within the time-of-harvest treatments (ToH) areas. Biomass amounts with shared letters are not significantly different. Error bars indicate the standard error. **(Bottom)** Sets of bars showing stem and foliage biomass for individual cultivars as labeled on the rightmost group.

### 3.3. Implications of Early Harvests on Biomass Production and Regrowth

One of the concerns with growing season harvesting is that willow aggressively produces coppice regrowth after cutting, interrupting photosynthetic activity and requiring the use of stored energy reserves to produce new shoots that may not harden sufficiently for winter. If new stems die off over the winter, this material, typically, is included in the next harvest and the plant has to produce new shoots again the following growing season. The mean combined biomass production for the 2007 and 2011 harvests ranged from approximately 80 Mg ha$^{-1}$ for harvests that occurred in June/August, to over 100 Mg ha$^{-1}$ for harvests in April (Figure 7); harvest timing was the only significant effect (Table 3). On closer inspection, despite a substantial increase in biomass production in the second rotation (Figure 6) compared to the first rotation (Figure 2), the loss in production and subsequent reduction in revenue may be considerable for harvests conducted between June and August; almost a 20% reduction compared to a dormant season harvest (Figure 7). Furthermore, the September timing had first rotation harvest yields comparable to the dormant seasons, but had the poorest second rotation yields, which suggests carbohydrates were expended on new growth that was unable to harden and survive through the winter.

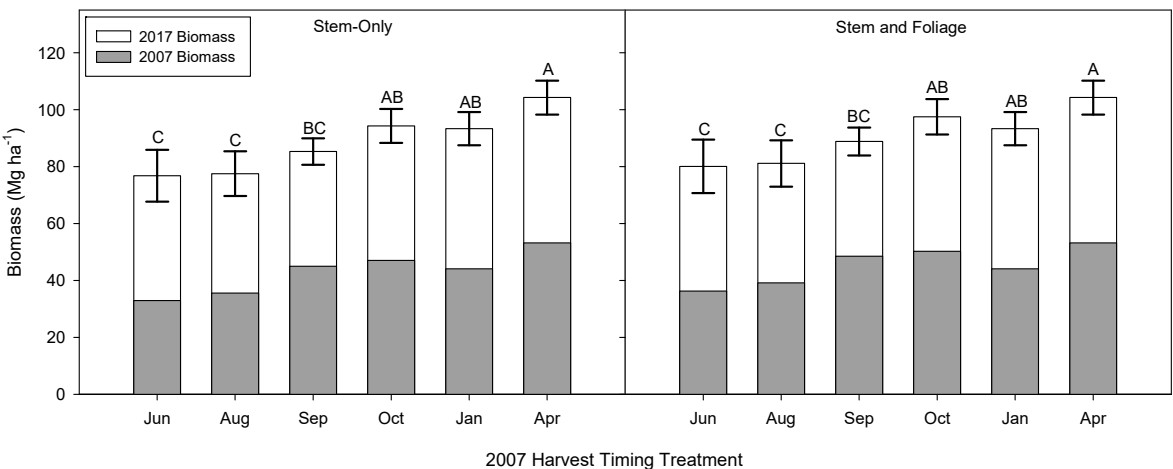

**Figure 7.** Combined biomass production (mean ± SE) for both 2007 and 2011 with and without foliage. Letters indicate significant differences within each graph only for combined biomass.

Harvest timing had a significant impact on the total biomass production over two rotations. Total biomass production over two rotations was significantly lower for harvest in June and August compared to October, January and April, indicating that harvesting during the growing season negatively impacts biomass production compared to dormant-season harvests (Figure 7). The lower production in the June and August harvests in the first rotation was still reflected in the second rotation, despite the higher GDD for the June and August harvests in the second rotation, which also translated to greater biomass production. The harvest in September was higher than June and August and lower than October and January, but none of these differences were significant. Total biomass production over two rotations was greatest for the April harvest and this was significantly greater than June August and September. The differences in total production during the dormant season (October, January and April) were small and not significant. The end result over two rotations was a decrease in production for harvests that occurred during the middle of the growing season in June and August.

In a comprehensive study comparing first- and second-rotation yields, when willow was harvested in the dormant season only, there was a propensity for second rotation yields to increase relative to the first rotation for a number of shrub willow cultivars across a range of sites [45]. This effect was stronger for willow with a lower yield in the first rotation. In this study, the mean first rotation yields in midsummer 2007 were 7.3 Mg ha$^{-1}$ year$^{-1}$ in June and 7.6 Mg ha$^{-1}$ year$^{-1}$ in August, and increased to 13.1 and 13.2 ha$^{-1}$ year$^{-1}$, respectively, in the second rotation. On the other hand, production for harvests in September, October, January, and April did not present significant changes up or down for the second rotation; however, they did show significant increases in their yield, given the shorter length of the second rotation (approximately 4 years) compared to the first (approximately 5 years). Sleight et al. [45] reported that when first rotation yield is between 9.4 and 12.9 Mg ha$^{-1}$ year$^{-1}$, the probability of increasing yield in the second rotation is <50%. First rotation yields in the current study for September, October, January, and April ranged from 9.5 to 10.6 Mg ha$^{-1}$ year$^{-1}$. Second rotation yields in our study increased by 78% for June and August, 28% for September, 52% for October, 87% for January, and 43% for April. Still, despite significant increases, the willow harvested during the growing season (June and August) did not match the yields obtained by the plants harvested during the dormant season, even though they had the equivalent of an extra growing season (in growing degree days (GDD); Table 2), in comparison to dormant-season harvests.

The effects of harvest timing on the growth of shrub willow, and other coppice species, have been studied [30–33,37,51]; despite this, gaps in understanding the impact of harvesting season on the plants' growth remain. Certain commonalities exist from previous

results; lower plant growth and development were observed when harvested during the growing season. The uncertainties about the harvest season effect are attributed to a variety of reasons, e.g., (1) possible lower root carbohydrate reserves when harvesting during summer, (2) frost damage of newly regenerated and immature shoots harvested late in the growing season, or (3) a limited nitrogen reserve supply for regrowth [30,36,37]. Our results showed similar growth responses to those found in previous studies. However, we cannot confirm any of these previous attributions. Woody nitrogen concentration in this study showed a significant increase from the growing season (3.3 to 3.5 mg $kg^{-1}$) to the dormancy season (4.1 to 4.5 mg $kg^{-1}$), with a significant decrease in early spring (3.4 mg $kg^{-1}$), during bud burst [20], similar to the observations made by other studies [26,36]. Although we did not study the specific dynamics and translocation of nitrogen, carbohydrates or other compounds in the plant, or evaluate the plants for frost damage, significant reductions and increases in leaf and woody nitrogen pools were observed as the plants approached the dormancy stage. Nonetheless, the reduced nutrient reserves observed during the growing season harvests (June, August, and September) are a plausible reason for the effect that harvest date had on biomass production and yield.

A caveat to the biomass concerns are that these plots were hand harvested and that large scale machine harvests may alter the results because not all the material is removed from the field. Hand harvesting is typically less aggressive and the collection of material very thorough. When SRWC are mechanically harvested, machines fail to collect some proportion of the material. For example, Berhongaray et al. [22] reported that poor efficiency can result in almost a quarter of the material being left behind; however, that study relied on exceptionally small sample areas. Eisenbies et al. [21] found that unrecovered biomass ranged between 6 and 8 percent for leaf-off harvests; however, it was on a limited number of plots. In some follow up work, a more extensive set of data from harvests during the growing season suggested that the amount of dropped material was under <9% of the harvestable biomass for 90% of the harvested area, <23% for 5% of the area, and <35% for 4% [19]. One percent or less of the area may have exceptionally high quantities of dropped material due to spillage of chips from the wagon or spout. However, harvesting systems are still under development and experience among operators is increasing so the percentage of dropped material has the potential to be reduced in the future. This material that is left on site may have an impact on soil nutrient and carbon status and plant growth in subsequent rotations that is not reflected in this hand harvested study. Another issue with growing season harvests is that foliage is not a desirable component for many end users [24,52,53]; future harvesting systems may incorporate wood/foliage separation in order to improve feedstock quality, especially if it improves feedstock grade and price, or offsets costs.

The results of the ToHxC interaction for total biomass production were highly influenced by the higher performance of cultivar SX-67 (Figures 2 and 6). It appeared to have high biomass production potential over both rotations, regardless of harvest timing. Additionally, cultivar 9882-25 did not show a strong response to harvest timing, but had relatively low total biomass production. Meanwhile, cultivars 9870-40 and 9871-41 both displayed a clearer harvest timing effect. These two distinct cultivar groups suggest, along with potential yield, that certain cultivars may be tolerant of an expanded harvesting season, while other cultivars are sensitive to growing season harvests. This result deserves further study in the future as the climate changes and the opportunities for only dormant-season harvests decline in the region.

### 3.4. Implications of Early Harvests on Nutrient Removal

Although NR is a concern, it is not currently a primary consideration in the harvest scheduling for operations in New York state due to weather windows and ground conditions. As discussed, the concerns over short rotations and frequent harvests of shrub willow principally relate to the quantity of nutrients removed, the potential impact on soil nutrient content, and the long-term productivity of the crop. This issue has been studied for dormant-season harvests [54–57]; the study of how the timing of harvest affects NR has

been limited, with most of the research performed on nutrient concentration, allocation, and translocation in the shrub willow biomass [26,58–60]. Our results indicate there are potential effects of harvest date on total removal (stem and foliage) of N and K, which may be relevant to decisions on harvest timing. Plants harvested in October had the highest total N and P, while K removal was higher in June and August. During the growing season, a high proportion of the nutrients are located in the foliage tissue; a portion of these nutrients are translocated into the shoots and root system in the fall [26,58] (Figures 3–5), which explains the higher removal observed during October and January in the stemwood. The timing in this study suggests that the window for minimum NR would be early spring. Alternatively, if it were possible to remove foliage during operations, a harvest late in the growing season could result in the most nutrients left on site, but also with low impact on the amount of biomass harvested. Additional study would be needed to identify this window.

NR was impacted by the interaction of the timing of the harvest and the different cultivars in this trial, having potentially interesting implications for the deployment and management of willow crops. Among the cultivars studied, SX67 resulted in consistently high total biomass and annual yield, but some nutrients had comparatively high NR across the harvest dates. Fabio et al., 2017 [61], studied the contributions of genotype and environment on shrub willow biomass composition, observing a high influence of environment as well as genotype–environment interaction on yield, and concluded that the selection of genotypes and growing environment could be implemented to increase biomass production. Their results can help explain the significant differences observed between SX67 and the other cultivars in our study, where environmental conditions could have been favorable for SX67 growth, compared to the other cultivars and regardless of the harvest date. Fabio et al. [61] also found two SX cultivars (SX61 and SX64), which belong to the same diversity group as SX67 (Table 1), to be stable and high yielding across a range of environmental conditions.

Alternatively, the cultivars 9870-40 and 9871-41, which had had comparable biomass production to SX67 from the September harvest (Figure 2), also appeared to have comparatively lower NR (Figures 3–5). For the macro-nutrients, 9870-40 and 9871-41 had approximately 10 kg N ha$^{-1}$ year$^{-1}$ and 2.5 kg P ha$^{-1}$ year$^{-1}$ lower exports; the difference for K only applied to cultivar 9871-41. While the cultivar SX67 was shown to be resilient to an expanded harvesting window in terms of biomass production, these results suggest that cultivar selection could also consider NR relative to harvest yield as selection criteria. As mentioned previously, if technology were introduced that separated wood and foliage during harvesting and redistributed this material on site, that could offset some removal, but the differences between cultivars in terms of foliage is not as notable.

Considering harvest date effects on biomass production and NR, the selection and deployment of different cultivars could be decided by best matching phenotypes to specific site conditions; however, this is based on a set of four cultivars and one site only. In this case, cultivar SX67 would be an example of a strong candidate to be deployed in sites where leaf-on harvests might be required for some portion of the life of the crop, ensuring high yield in the following rotation; however, SX67 also showed a variable and high NR across harvest dates. Hence, at a site such as this, deploying a combination of SX67 and 9871-41 (which resulted in variable yielding and low and variable NR across the harvest dates) could be beneficial both for the overall yield and NR rates on the site. Clearly, such management decisions would need to be tailored for regions, sites, and cultivars [38]. Assuming harvests occur from August to October (as observed in commercial sites in NY, given poor site conditions in the fall and winter seasons) a combined planting of SX67 and 9871-41 would have mean yields of 13.8 Mg ha$^{-1}$ year$^{-1}$ and removals of 69.7 kg N ha$^{-1}$ year$^{-1}$, 9.9 kg P ha$^{-1}$ year$^{-1}$, and 42.4 kg K ha$^{-1}$ year$^{-1}$. In contrast, a deployment of cultivars with characteristics similar to SX67 only, would ensure higher yields (15.5 Mg ha$^{-1}$ year$^{-1}$), but likely result in higher NR rates (80.5 kg N ha$^{-1}$ year$^{-1}$, 12.0 kg P ha$^{-1}$ year$^{-1}$, and 51.3 kg K ha$^{-1}$ year$^{-1}$).

A careful selection of cultivar and growing environment, as explained by Fabio et al. [61], could ensure higher yields over rotations, as well as similar results when harvesting on different dates and in different seasons of the year. Hence, by considering cultivars with lower variation both in yield and NR, a wider harvesting window could be supported, ensuring the biomass production of subsequent rotations and facilitating the nutrient management practices. The addition of on-site leaf separation could also decrease the amount of nutrients removed from the site, giving managers additional choices in balancing biomass production and NR. However, a wider array of cultivars and sites should be explored to confirm the patterns observed, since a limited suite of cultivars and only one site were used in this study.

*3.5. Implications of Harvest Dates for Commercial Operations*

This study suggests that harvesting during the plant's dormancy stage (late fall, winter, and early spring) will promote higher biomass production. If nutrient retention is desired, the optimal times may be late September to early spring before leaf out to limit export from the site via harvested biomass. The obvious limitation of spring harvests is the ground conditions, which would require equipment that is suited for operating with low ground pressures [24]. Best practices for shrub willow management for many years recommended that harvesting take place during winter months, after leaf fall has already occurred [23], based, in part, on studies indicating higher biomass production and shrub willow growth when the harvest is performed during the plant's dormancy stage [30,32,33,37]. However, commercial shrub willow harvest operations in NY are being extended into the mid-late growing season because following best practices has proven challenging due to weather and ground conditions on large sites. The future expansion of biorefineries that require a year-round supply of biomass may also prompt an expansion of the harvesting window because of the dry matter loss associated with willow that is stored for three to six months depending on the time of harvest. The tradeoff between differences in yield with more frequent harvests over the entire year and losses in both quality and quantity of biomass that is stored for longer periods of time from dormant-season only harvests needs to be assessed and incorporated into the analysis that includes different conversion processes and end products.

Shrub willow crops are commonly planted on marginal agricultural land in NYS [6]. The term "marginal land" has different meanings in various settings, but here refers to land at the margins of profit, where potential economic returns are at a breakeven point with production costs [62], which was recently characterized as socially marginal land when externalities are included [63]. These lands generally have use restrictions, caused by slope, elevation, depth, soil texture, internal drainage, fertility, and/or remoteness. In the northeast US, common limitations for this land are most often related to the hydrology, which results in seasonal saturation or near saturation [6]. Hence, it has been observed that the operation of heavy machinery on these lands during wintertime requires frozen ground. If snowfalls come before the ground freezes to support heavy equipment, access to the site may be hindered for the entire season, or operating costs become prohibitive.

According to this study, harvesting during August will result in significantly lower total biomass production and yield compared to fall or winter harvests. Total biomass production for the August harvest date resulted in 77.5 Mg ha$^{-1}$, while in October the total biomass production was 94.3 Mg ha$^{-1}$. Considering a wet biomass price at the plant gate in US dollars (USD) 30.5 Mg$^{-1}$ [64], we could estimate a gross revenue of USD 4584.2 ha$^{-1}$ after two rotations if harvesting during August and USD 5236.9 ha$^{-1}$ if harvested during October (Table 6). Still, if the results of this research are considered, and the harvest is performed during April, it would result in a total of USD 5731 ha$^{-1}$ after two rotations. These results, however, do not consider other costs or incomes in the system, only the economic return generated by selling the biomass. As already mentioned, harvesting during rainy or snowy periods could increase the harvesting costs, add delays to harvesting with small amounts of rainfall because evapotranspiration is so low during

the dormant season, or prohibit the harvest from happening altogether. In addition to differences in biomass production, recent analysis of willow harvests has shown that harvester throughput from leaf-on harvests in dry ground conditions (29.7 Mg h$^{-1}$) are 59% lower than leaf-off harvests on dry ground (71.8 Mg h$^{-1}$). This would increase harvesting costs and reduce the profitability of leaf-on harvests [24]. In addition, leaf-off harvest throughput in wet conditions (42.4 Mg h$^{-1}$) was 41% lower than when in dry conditions. Despite higher biomass production and gross return generated in April, spring snow melt could contribute to soil water saturation, resulting in site conditions not ideal for operating harvesting equipment and increasing harvesting costs, which could lead to lower net revenue compared to other months, when harvesting conditions are ideal. The tradeoffs between losses in production and the increased costs associated with trying to harvest on these poorly drained sites during the dormant season is another factor that needs to be incorporated into decisions about the timing of harvesting operations.

**Table 6.** Gross revenue from willow biomass depending on total biomass production for each harvest date. Price of biomass at gate is considered at USD 30.5/Mg using the EcoWillow 2.0 Cash Flow Model [52]. No additional costs or incomes are considered.

| Harvest Date | First Rotation | | Second Rotation | | Gross Revenue over Two Rotations |
|---|---|---|---|---|---|
| | Biomass Wet | Gross Revenue * | Biomass Wet | Gross Revenue * | |
| | Mg ha$^{-1}$ | USD ha$^{-1}$ | Mg ha$^{-1}$ | USD ha$^{-1}$ | USD ha$^{-1}$ |
| June | 65.8 | 2006.9 | 82.6 | 2519.3 | 4526.2 |
| August | 71.2 | 2171.6 | 79.1 | 2412.6 | 4584.2 |
| September | 76.3 | 2327.2 | 76.0 | 2318.0 | 4645.2 |
| October | 82.5 | 2516.3 | 89.2 | 2720.6 | 5236.9 |
| January | 80.2 | 2446.1 | 93.0 | 2836.5 | 5282.6 |
| April | 91.7 | 2796.9 | 96.2 | 2934.1 | 5731.0 |

* Gross revenue = biomass as received (Mg) = USD 30.5 Mg$^{-1}$.

NR presented a pattern inverse to biomass production but similar to annual yield, in which higher removals were observed for harvest dates during fall (October), followed by summer (June, August, and September) or winter (January) harvests, and generally lower in spring (April), especially for N and P, which are probably the ones that most often limit plant's growth and receive attention [16,65]. Additionally, soil N and P levels have been shown to decrease significantly after several shrub willow rotations [20]. Our results indicate that the ideal season to perform harvest would be early spring prior to leaf out. Harvesting during early spring would then ensure higher yields with a nutrient export that is lower than harvesting at other times of the year. However, a considerable amount of the nutrients removed during summer and fall harvest dates are present in the leaves, while no leaves are removed during the winter and spring harvests. Considering only the nutrients removed in the woody biomass, we observed that summer harvest (June, August, and September) removed similar amounts as spring harvest (April). We assumed that all leaf material (entire crown of the plant) was harvested during leaf-on stages; however, as previously mentioned, a high proportion of the foliage (data not available) remains on the site to decompose after a commercial, mechanized harvest.

Another consideration is the potential to improve the existing single pass cut and chip harvesting system through modifications to facilitate the separation of leaves and increase the amount of this material returned to the site or to increase the harvester's flotation to operate during wet soil conditions and avoid leaf-on harvests. This would reduce NR and improve soil conditions and the quality of the biomass that is collected for conversion to renewable energy products [66] and could help growers expand the time of year for harvesting.

Commonly, results of NR are obtained from hand harvests and field trials. Observations of commercial shrub willow harvesting operations have shown that nutrient rich woody and leaf biomass is left on the site. Soil N and P levels have been noted to decrease after several rotations [20], which could possibly have impacts on the crop's long-term productivity. For instance, if the efficiency of the harvester is between 7 and 15% of the total standing biomass, that would represent 20–35% of the total nutrient content in the woody above-ground willow biomass [20]. This material would remain on site and offset nutrient losses as they decompose. More research is needed into commercial harvesting operations to determine the amount of dropped biomass for harvests at different times of the year (both woody and leaf) and the nutrient content in this biomass, as well as to observe how these operations impact the soil's nutrient levels and the crop's long-term productivity.

## 4. Conclusions

The total biomass production and NR results from this study support the common recommendation to harvest willow after leaf drop whenever possible. To ensure higher biomass production over multiple rotations, shrub willow crops in NY should be harvested during leaf-off stage when possible. However, site and climatic limitations have forced commercial growers in NY to start harvesting operations in the middle to late growing season, rather than the dormant season. Harvesting during the growing season can affect the next rotation's biomass production and possibly remove higher amounts of nutrients from the site. These impacts could be mitigated by the deployment of cultivars whose biomass production is not compromised by leaf-on harvest. Additionally, the development of methods to separate foliage from woody biomass during harvesting operations that could contribute to the retention of foliage on site is needed. Improving the harvesting equipment's ability to operate in wet soil conditions could allow for a greater proportion of harvesting to occur during the dormant season, reducing the negative impacts on production and nutrient losses.

The different biomass production and nutrient removal results between the cultivars and harvest dates suggests that nutrient management and harvesting recommendations will not be effective for all cultivars and sites. The influences of environment and genotype on yield have been observed before, and our results indicate a similar effect of the interaction between cultivar and harvest date on both biomass production and NR. Therefore, harvesting and nutrient management guidelines should consider site and cultivar characteristics to ensure high yields and maintain soil nutrient conditions over multiple rotations.

Further research is needed to understand the importance of biomass (both leaf and woody) that is dropped on site during mechanized harvesting operations. Harvesting system improvements that focus on separating foliage and woody biomass to increase the proportion of woody biomass removed while retaining the nutrient-rich foliage and non-merchantable biomass (small twigs and tops of plants) on the site could reduce the nutrient-removal impact of both leaf-on and leaf-off harvests and potentially improve the quality of the harvested biomass.

**Author Contributions:** Conceptualization, T.A.V. and D.P.D.S.; methodology, D.P.D.S., M.H.E. and T.A.V.; formal analysis, M.H.E., D.P.D.S. and T.A.V.; investigation, D.P.D.S., T.A.V. and M.H.E.; resources, T.A.V.; data curation, D.P.D.S.; writing—original draft preparation, D.P.D.S.; writing—review and editing, M.H.E. and T.A.V.; visualization, M.H.E. and D.P.D.S.; supervision, T.A.V.; project administration, T.A.V.; funding acquisition, T.A.V. and M.H.E. All authors have read and agreed to the published version of the manuscript.

**Funding:** A portion of this research is part of the MASBio (Mid-Atlantic Biomass Consortium for Value-Added Products) project, which is supported by the Agriculture and Food Research Initiative Competitive Grant no. 2020-68012-31881 from the USDA National Institute of Food and Agriculture. This research was also made possible by funding under award #EE0006638 from the U.S. Department of Energy Bioenergy Technologies Office.

**Data Availability Statement:** The authors are willing to share data and collaborate with potential partners upon request.

**Acknowledgments:** Longer term field studies are often possible only with the engagement and assistance from a number of people. We are grateful for many people who helped with various aspects of this work. Ben Ballard was instrumental in the initial trial design, layout and initial data collection. Rebecca Almond, Mark Appleby, Jen Ballard, Ken Burns, Eric Fabio, Godfrey Ofenzu, and Ruth Owens managed the data collection, sample processing and data management at different stages in this trial. Different teams of students, including, Gabe Kellman, Ryan Newby, Dan Quinn, Stephanie Teale, Lucas Wachob, Jed Walsh and others were instrumental in completing field and lab work.

**Conflicts of Interest:** The authors declare no conflict of interest.

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
