# Peer review of "Growing Season Harvests of Shrub Willow (Salix spp.) Have Higher Nutrient Removals and Lower Yields Compared to Dormant-Season Harvests"

_forests, doi:10.3390/f13111936_

Round 1

Reviewer 1 Report

The paper entitled:" Growing Season Harvests of Shrub Willow (Salix spp.) Have Higher Nutrient Removals and Lower Yields Compared to Dormant Season Harvests ", discusses the possible influence of leaf-on harvesting on willow regrowth, nutrient exports, and its implications for short rotation willow management, the case of central NY. While the study's importance is high and the results obtained during this investigation are in accordance with the conclusions drawn by the authors, the paper methdology still, somehow, raising doubts, the others must clarify the absence of any experimental dispositive (set-up) in their work (split-plot ?).

Lines 155-156: "While this approach does release plants in subsequent plots, the purpose was to ensure equitable lighting for sprouts that emerge from willow stools soon after cutting", how is that possible. please clarify

Author Response

Reviewer 1

Comments and Suggestions for Authors

The paper entitled:" Growing Season Harvests of Shrub Willow (Salix spp.) Have Higher Nutrient Removals and Lower Yields Compared to Dormant Season Harvests ", discusses the possible influence of leaf-on harvesting on willow regrowth, nutrient exports, and its implications for short rotation willow management, the case of central NY. While the study's importance is high and the results obtained during this investigation are in accordance with the conclusions drawn by the authors, the paper methdology still, somehow, raising doubts, the others must clarify the absence of any experimental dispositive (set-up) in their work (split-plot ?).

In the statistical methods (section 2.4) we state that the study was analyzed as a strip-strip plot.  To address this comment we state again that we utilized a strip-strip plot

Lines 155-156: "While this approach does release plants in subsequent plots, the purpose was to ensure equitable lighting for sprouts that emerge from willow stools soon after cutting", how is that possible. please clarify

We’ve eliminated the sentence because the point being made is redundant to the information provided in the preceding sentences.  It seems to have caused confusion for this reader, so we have eliminated it.

Reviewer 2 Report

Biomass supplement is an important part for a bioenergy and bioproducts industry. Short rotation woody crops (e.g. willow) are considered potential biomass feedstocks to replace the fossil fuels, especially in the U.S. where traditional energy supply dominates. Therefore, a reliable supply and demand for biomass must be guaranteed. The manuscript studied the potential impact leaf-on harvesting on willow regrowth and nutrient export, and its implications for short rotation willow management. This topic fits to the scope of the Forests Journal. However, the manuscript it shows several significant issues with data analysis and figures, and the expression of results and text format also need further improvement.

The foremost criticism I have of this manuscript is the data analysis, which the replications not be include as a random factor in the mixed model for estimate the effects of time-of-harvest treatments (ToH), cultivars (C), and ToHxC interactions on total biomass production, nutrient removal, etc. Because cultivar SX67 showed higher variation between individuals (replications) compared to other three cultivars (for example in Figure 3). Such high variation should be explain and may affect the final results. On the other hand, the figures are not scientific and puzzled. For example, in Figure 2 of the bottom one, the error bar for the total biomass (upward) is incorrect, which its starting point varied.

The second criticism I have is the expression of results and text format are not well organized, lack of logic and main point, which need great improvement. The expression in the manuscript is redundant, looks like a dissertation part rather than a paper (L708), especially in the Results and Discussion part. We suggest some parts could be merged, for example merge 3.1.2, 3.1.3 and 3.1.4 into one. Besides that, for the text format, there is an extra space at the beginning of a sentence throughout the whole paper. More time is needed to polish this paper.

In the Abstract part, author should tell the reader whether the influences of harvest timing among four willow cultivars were significantly different? Which cultivar had the highest resilient to the harvest? Moreover, beside the harvest yield comparison between growing season and dormant season, it need to further discuss cultivars’ performance throughout the growing season (Jun/Aug/Sep), and try to give out more specific month information for harvest. Yet, these results now are missing in the Abstract part and Discuss part, which are important and should be clarify.

Specific comments:

Abstract

L10, Please note the spp. do not need to be in italics. Same in L29

L14-16, Please clarify the method more clearly, help reader to understand the meaning of first rotation harvest and second rotation harvest.

L24, Better to delete one “Short rotation woody crops” or “short rotation coppice”; because they have similar meaning.

Introduction

L98-107, Better give out your hypothesis, which will make the paper more readable.

Results

L222, Please note it should be P=0.0068 not P<0.0068.

L234, Please remove the extra punctuation.

L258-263, Better to discuss the influence of foliage to the result rather than emphasis its undesirable for user.

L274-275, Please explain why “Significant effects of harvest date, cultivar, and ToHxC were observed on total N removal”. From the Figure 3, we can’t obtain this information. Instead, it observed cultivar effects for total N do not significant (P=0.0792).

L322, Please indentation.

L383, Please check whether it is Figure 3 or Figure 6?

L390, Please change the X-axis “2007 Time of Harvest Treatment” into “2007 Harvest Timing Treatment”.

L398, Please discuss the influence of high individual variation (error bar in Figure 6) in SX67 when make the conclusion SX67 may be more resilient to the expanded harvest seasons.

L494, There is no Figure 9 in your paper. Please check it.

Reference

R8, 10, 34, 45, 49, the plant Latin name should be in italics.

Author Response

Reviewer 2

Comments and Suggestions for Authors

Biomass supplement is an important part for a bioenergy and bioproducts industry. Short rotation woody crops (e.g. willow) are considered potential biomass feedstocks to replace the fossil fuels, especially in the U.S. where traditional energy supply dominates. Therefore, a reliable supply and demand for biomass must be guaranteed. The manuscript studied the potential impact leaf-on harvesting on willow regrowth and nutrient export, and its implications for short rotation willow management. This topic fits to the scope of the Forests Journal. However, the manuscript it shows several significant issues with data analysis and figures, and the expression of results and text format also need further improvement.

The foremost criticism I have of this manuscript is the data analysis, which the replications not be include as a random factor in the mixed model for estimate the effects of time-of-harvest treatments (ToH), cultivars (C), and ToHxC interactions on total biomass production, nutrient removal, etc. Because cultivar SX67 showed higher variation between individuals (replications) compared to other three cultivars (for example in Figure 3). Such high variation should be explain and may affect the final results. On the other hand, the figures are not scientific and puzzled. For example, in Figure 2 of the bottom one, the error bar for the total biomass (upward) is incorrect, which its starting point varied.

We decided to use a Split-Split analysis because although he months are arguably fixed, because of the layout of the study, where we applied the ToH treatment from south to north, we were potentially subject to topographic or drainage effects.  Thus, we used the strip strip approach which is more conservative. 

The choice for error bars being based on arithmetic means is stylistic, not scientific.  If we used the model errors, the bars would all be the same and provide little insight about the individual variability.  Ecological studies are fraught with such frustrations in presenting material; I’m not sure there’s ever a right choice.  This is the one we made. 

To address this comment, we revised the headings to emphasize they are one-way error bars and made it more explicit what each bar represents.  Reviewer 4 suggested adding identical letters even when none of the differences are significant.  Although this is a little unconventional, it may help alleviate the confusion about error bars.  So we have done that as well.

With regards to highlighting specific cultivars, we addressed the expected issue with SX-67 using preplanned contrasts.  That was the best tool at our disposal for teasing out these patterns.

We made the explanation of the error bars in the figure captions more explicit.

The second criticism I have is the expression of results and text format are not well organized, lack of logic and main point, which need great improvement. The expression in the manuscript is redundant, looks like a dissertation part rather than a paper (L708), especially in the Results and Discussion part. We suggest some parts could be merged, for example merge 3.1.2, 3.1.3 and 3.1.4 into one. Besides that, for the text format, there is an extra space at the beginning of a sentence throughout the whole paper. More time is needed to polish this paper.

This paper had gone through many internal revisions over its organization.  Based on earlier reviews, readers were confused by the difference between first and second rotation biomass production (as this reviewer even requests clarification below) and harvest timing.  Thus, we settled on first-rotation effects (biomass and nutrients), and second rotation effects (biomass only), then discussing overall implications.  We also tracked more nutrients in earlier iterations. As a result I believe the reader has picked up on some residual statements that affect the flow of the conversation (including a improper figure reference).  We have tried to carefully do a general revision for the sake of continuity.

With regards to sections 3.1.2, 3.1.3, and 3.1.4., we agree that the section numbering was not helpful.  It is also related to the overhaul we made on organization and was meant to make subjects easier to find, but we were not effective.  We eliminated the subsection headings.

In the Abstract part, author should tell the reader whether the influences of harvest timing among four willow cultivars were significantly different? Which cultivar had the highest resilient to the harvest? Moreover, beside the harvest yield comparison between growing season and dormant season, it need to further discuss cultivars’ performance throughout the growing season (Jun/Aug/Sep), and try to give out more specific month information for harvest. Yet, these results now are missing in the Abstract part and Discuss part, which are important and should be clarify.

We are limited by the journal requirements for abstract length and there simply isn’t additional room for everything the reviewer requests.  We made the suggested wording correction.  With regards to the performance of specific cultivars, we wish to downplay any implied promotion of a given cultivar since their performance can be quite dynamic from site to site.  We are trying to be conservative about conclusions drawn relative to specific cultivars

Specific comments:

Abstract

L10, Please note the spp. do not need to be in italics. Same in L29

Thank you

L14-16, Please clarify the method more clearly, help reader to understand the meaning of first rotation harvest and second rotation harvest.

Rotation is a common term in forestry and silviculture.  There is no room to define such basic terms in the abstract.  It is already implied by the harvest-cycle phrase earlier in the abstract

L24, Better to delete one “Short rotation woody crops” or “short rotation coppice”; because they have similar meaning.

Agree that it was redundant

Introduction

L98-107, Better give out your hypothesis, which will make the paper more readable.

We added the sentences " As such the hypothesis is that first, the time of year that a harvest occurs will affect nutrient removals based on the amount and type of plant materials that are removed with the harvest.  Second, harvest timing will affect subsequent regrowth.  Third, differences may be cultivar dependent."

Results

L222, Please note it should be P=0.0068 not P<0.0068.

Corrected

L234, Please remove the extra punctuation.

Thank you

L258-263, Better to discuss the influence of foliage to the result rather than emphasis its undesirable for user.

The sentence was out of place in that part of the paragraph.  We moved it to an an earlier point in the paragraph to serve as a segue

L274-275, Please explain why “Significant effects of harvest date, cultivar, and ToHxC were observed on total N removal”. From the Figure 3, we can’t obtain this information. Instead, it observed cultivar effects for total N do not significant (P=0.0792).

The ToHxC interaction was significant 0.0062 (Table 4).  Reviewer 4 had a similar comment about figure 3.  As you can imagine adding letters to the figures was problematic, which is another reason we utilized contrasts.  The important ToHxC interactions are highlighted in the rest of the discussion in this and subsequent paragraphs.

L322, Please indentation.

Thank you.

L383, Please check whether it is Figure 3 or Figure 6?

Thank you.

L390, Please change the X-axis “2007 Time of Harvest Treatment” into “2007 Harvest Timing Treatment”.

Thank you

L398, Please discuss the influence of high individual variation (error bar in Figure 6) in SX67 when make the conclusion SX67 may be more resilient to the expanded harvest seasons.

Regardless of the increased variation, the overall mean is high, so as a crop, its overall performance is higher.  SX67 seems to jump off the blocks in terms of biomass production, whereas two of the cultivars steadily gain over the course of the season.  We have to be careful about significance hunting, which is why we try to temper our conclusions by not focusing on specific cultivars, only that these patterns should be watched.

We altered the paragraph slightly to address this comment.

"... What is particularly interesting is these results suggest that certain cultivars, in this case SX67, although it has higher variability its higher mean suggests it may be more resilient to the expanded harvest seasons that may be necessary to support large-scale commercial willow system. Regardless of when SX67 was harvested in the first rotation, second rotation yields were high and fairly consistent. Both high yield, and a tolerance for early-season harvesting may be a necessary selection criteria for cultivar deployments.  However, matching cultivars to favorable sites can be a challenge; the purpose of this paper is not to promote specific cultivars.  "

L494, There is no Figure 9 in your paper. Please check it.

This was a holdover from the internal revision where we reduced the manuscripts scope.  The reference has been corrected.

R8, 10, 34, 45, 49, the plant Latin name should be in italics.

Thank you

Reviewer 3 Report

Over all the authors have written the manuscript very well. However the title of the article seems not good. Try to modify that one.

Add some latest references. 

Conclusion should be rewritten.

Author Response

Reviewer 3

Comments and Suggestions for Authors

Over all the authors have written the manuscript very well. However the title of the article seems not good. Try to modify that one.

The criticism is too general and provides no guidance.  No comment from the other reviewers.  Our preference is to have the title tell some of the story

Add some latest references.

25% of the references are since 2015 and over half the references are since 2010.  Three quarters are since 2000.  We already have over 69 citations.  Without additional guidance this is a difficult comment to address within a reasonable timeframe for submitting our revision.

Conclusion should be rewritten.

We have made significant edits to the entire conclusions section.

Reviewer 4 Report

Dear Authors,

The manuscript submitted for review entitled: “Growing Season Harvests of Shrub Willow (Salix spp.) Have Higher Nutrient Removals and Lower Yields Compared to Dormant Season Harvests”: De Souza Daniel, Mark H. Eisenbies, Timothy Volk I have read with great interest. The research undertaken by you is a good fit with the publications presented in the “Forests” Journal. At the moment I rate your work relatively highly. The authors recognize that Salix species harvested during the growing season have higher nutrient removal compared to those harvested during the dormant season, even though the harvest of willow during the growing season is characterized by lower yields. The presented research documents the amount of uptake of minerals from the soil through the stems, leaves, and the stems and leaves. The size of nutrient removal was calculated based on the chemical composition of the yield and the size of the yield. Well-structured conclusions indicate greater benefits of harvesting stems without leaves in the Dormant Season, which reduces nutrient losses from the sites where the Salix crops are located. Economic calculations are also an interesting element of the manuscript.

I found a few inaccuracies which require corrections. I have included all my comments in the pdf file attached as the reviewer’s comments

The most important observations that you should take into consideration are:

Row 78 and others: Authors may consider introducing the abbreviation NR (nutrient removal) throughout the manuscript.

Row 190, 191: Please provide the apparatus used for N total determination (model, type, producer, city, country).

Row 193: Did authors forget about potassium determination? Could the authors provide the method and apparatus used for Potassium determination?

Row 231: If no significant differences were found, the same letters (A-A-A) should be marked on the bars in all ToH. OK?

Row 315: Please, move Figure 4 to section 3.1.3 Phosphorus

Row 370: Please, move Figure 5 to section 3.1.4 Potassium

Rows 705: Please use the MDPI layout as per the suggestions for the authors. Author Contributions: For research articles with several authors, a short paragraph specifying their individual contributions must be provided. The following statements should be used “Conceptualization, X.X. and Y.Y.; methodology, X.X.; software, X.X.; validation, X.X., Y.Y. and Z.Z.; formal analysis, X.X.; investigation, X.X.; resources, X.X.; data curation, X.X.; writing—original draft preparation, X.X.; writing—review and editing, X.X.; visualization, X.X.; supervision, X.X.; project administration, X.X.; funding acquisition, Y.Y. All authors have read and agreed to the published version of the manuscript.”

Please note other minor corrections in the attached pdf file

Yours sincerely

Reviewer

Author Response

Reviewer 4

Comments and Suggestions for Authors

Dear Authors,

The manuscript submitted for review entitled: “Growing Season Harvests of Shrub Willow (Salix spp.) Have Higher Nutrient Removals and Lower Yields Compared to Dormant Season Harvests”: De Souza Daniel, Mark H. Eisenbies, Timothy Volk I have read with great interest. The research undertaken by you is a good fit with the publications presented in the “Forests” Journal. At the moment I rate your work relatively highly. The authors recognize that Salix species harvested during the growing season have higher nutrient removal compared to those harvested during the dormant season, even though the harvest of willow during the growing season is characterized by lower yields. The presented research documents the amount of uptake of minerals from the soil through the stems, leaves, and the stems and leaves. The size of nutrient removal was calculated based on the chemical composition of the yield and the size of the yield. Well-structured conclusions indicate greater benefits of harvesting stems without leaves in the Dormant Season, which reduces nutrient losses from the sites where the Salix crops are located. Economic calculations are also an interesting element of the manuscript.

I found a few inaccuracies which require corrections. I have included all my comments in the pdf file attached as the reviewer’s comments

The most important observations that you should take into consideration are:

Row 78 and others: Authors may consider introducing the abbreviation NR (nutrient removal) throughout the manuscript.

Thank you for the suggestion

Row 190, 191: Please provide the apparatus used for N total determination (model, type, producer, city, country).

As stated, determinations were made at the lab at Penn State.  We do not have access to this information.

Row 193: Did authors forget about potassium determination? Could the authors provide the method and apparatus used for Potassium determination?

As stated, determinations were made at the lab at Penn State.  We do not have access to this information.

Row 231: If no significant differences were found, the same letters (A-A-A) should be marked on the bars in all ToH. OK?

Thank you for the suggestion.  Although atypical to include letters when there are no significant differences, this suggestion may help resolve the confusion over the error bars that reviewer 2 had.

Row 315: Please, move Figure 4 to section 3.1.3 Phosphorus

We eliminated the section subheaders.

Row 370: Please, move Figure 5 to section 3.1.4 Potassium

We eliminated the section subheaders.

Rows 705: Please use the MDPI layout as per the suggestions for the authors. Author Contributions: For research articles with several authors, a short paragraph specifying their individual contributions must be provided. The following statements should be used “Conceptualization, X.X. and Y.Y.; methodology, X.X.; software, X.X.; validation, X.X., Y.Y. and Z.Z.; formal analysis, X.X.; investigation, X.X.; resources, X.X.; data curation, X.X.; writing—original draft preparation, X.X.; writing—review and editing, X.X.; visualization, X.X.; supervision, X.X.; project administration, X.X.; funding acquisition, Y.Y. All authors have read and agreed to the published version of the manuscript.”

Converted

Please note other minor corrections in the attached pdf file

Corrections made

Yours sincerely

Reviewer

Round 2

Reviewer 1 Report

The authors have clarified all the addressed comments. 

Reviewer 2 Report

Just accept